# Maternal Iodine Status and Associations with Birth Outcomes in Three Major Cities in the United Kingdom

**DOI:** 10.3390/nu11020441

**Published:** 2019-02-20

**Authors:** Charles J. P. Snart, Claire Keeble, Elizabeth Taylor, Janet E. Cade, Paul M. Stewart, Michael Zimmermann, Stephen Reid, Diane E. Threapleton, Lucilla Poston, Jenny E. Myers, Nigel A. B. Simpson, Darren C. Greenwood, Laura J. Hardie

**Affiliations:** 1Department of Clinical and Population Science, Leeds Institute of Cardiovascular and Metabolic Medicine, School of Medicine, University of Leeds, Leeds LS2 9JT, UK; medcjps@leeds.ac.uk (C.J.P.S.); E.Taylor@leeds.ac.uk (E.T.); D.E.Threapleton@leeds.ac.uk (D.E.T.); D.C.Greenwood@leeds.ac.uk (D.C.G.); 2Leeds Institute for Data Analytics, University of Leeds, Leeds LS2 9JT, UK; C.M.Owen@leeds.ac.uk; 3Nutritional Epidemiology Group, School of Food Science & Nutrition, University of Leeds, Leeds LS2 9JT, UK; J.E.Cade@leeds.ac.uk; 4Faculty of Medicine and Health, University of Leeds, Leeds LS2 9JT, UK; P.M.Stewart@leeds.ac.uk; 5Laboratory for Human Nutrition, Institute of Food, Nutrition and Health, Swiss Federal Institute of Technology (ETH), 8092 Zürich, Switzerland; michael.zimmermann@hest.ethz.ch; 6Earth Surface Science Institute, School of Earth and Environment, University of Leeds, Leeds LS2 9JT, UK; S.Reid@leeds.ac.uk; 7Division of Women’s Health, Women’s Health Academic Centre, King’s College London, London SE1 7EH, UK; lucilla.poston@kcl.ac.uk; 8Maternal and Fetal Health Research Centre, Institute of Human Development, University of Manchester, Manchester M13 0JH, UK; Jenny.Myers@manchester.ac.uk; 9Division of Women’s and Children’s Health, School of Medicine, University of Leeds, Leeds LS2 9JT, UK; N.A.B.Simpson@leeds.ac.uk

**Keywords:** Iodine, pregnancy, birth weight, Insufficiency, SGA, preterm Birth

## Abstract

Severe iodine deficiency in mothers is known to impair foetal development. Pregnant women in the UK may be iodine insufficient, but recent assessments of iodine status are limited. This study assessed maternal urinary iodine concentrations (UIC) and birth outcomes in three UK cities. Spot urines were collected from 541 women in London, Manchester and Leeds from 2004–2008 as part of the Screening for Pregnancy End points (SCOPE) study. UIC at 15 and 20 weeks’ gestation was estimated using inductively coupled plasma-mass spectrometry (ICP-MS). Associations were estimated between iodine status (UIC and iodine-to-creatinine ratio) and birth weight, birth weight centile (primary outcome), small for gestational age (SGA) and spontaneous preterm birth. Median UIC was highest in Manchester (139 μg/L, 95% confidence intervals (CI): 126, 158) and London (130 μg/L, 95% CI: 114, 177) and lowest in Leeds (116 μg/L, 95% CI: 99, 135), but the proportion with UIC <50 µg/L was <20% in all three cities. No evidence of an association was observed between UIC and birth weight centile (−0.2% per 50 μg/L increase in UIC, 95% CI: −1.3, 0.8), nor with odds of spontaneous preterm birth (odds ratio = 1.00, 95% CI: 0.84, 1.20). Given the finding of iodine concentrations being insufficient according to World Health Organization (WHO) guidelines amongst pregnant women across all three cities, further studies may be needed to explore implications for maternal thyroid function and longer-term child health outcomes.

## 1. Introduction

Iodine is essential for the synthesis of the thyroid hormones triiodothyronine (T3) and thyroxine (T4) which regulate growth and metabolism [1]. Maternal iodine requirements are increased throughout pregnancy to support the thyroid hormone demands of the developing foetus [2]. The foetus is initially reliant on maternal thyroid hormones, but as the foetal thyroid begins functioning from 15–17 weeks gestation, it depends on the maternal iodine supply to maintain thyroid hormone production throughout the remainder of pregnancy [2]. Iodine requirements increase throughout gestation, and iodine deficiency during pregnancy is associated with a number of adverse outcomes for the child, including increased mortality, decreased cognitive performance and delayed physical development [1,3]. Severe deficiency during pregnancy can result in cretinism, a condition characterised by severe mental impairment, motor spasticity and deaf-mutism [1,3].

The introduction of iodine into livestock feed and the widespread adoption of iodophors in the dairy industry after the 1930s resulted in the reduction of overt signs of iodine deficiency in the United Kingdom (UK) population by the 1960s [4,5,6]. To date, the UK has not implemented an iodine fortification programme for commercial bread or salt production unlike countries such as Denmark and Australia [7,8]. Some authors have suggested that the use of iodophors in the dairy and bread industries in the UK has declined and been superseded by non-iodine alternatives, potentially reducing these sources of dietary iodine [1,9]. In addition, wider changes in dietary practice such as reduced milk and fish intake, the increased availability of dairy substitutes and an increase in vegetarianism may also be affecting iodine status in the UK population [9].

Although the UK adult population is considered iodine sufficient [10], recent studies have suggested that women may become deficient during pregnancy [11,12,13,14,15,16]. Few studies have provided an up-to-date measure of iodine status during pregnancy, nor have they attempted to link insufficiency with birth outcomes [12,13,14,15]. Additionally, these studies have been geographically limited, with many being set in single cities or regions in southern England [12,13,14,15,16]. Recent nationally-representative data have suggested that women of child-bearing age in the UK may be iodine insufficient, though this study specifically excluded pregnant and lactating women and did not assess birth outcomes [10].

We therefore aimed to assess iodine status in the three UK sites of the Screening for Pregnancy Endpoints (SCOPE) international birth cohort, and assess associations between maternal iodine status and birth outcomes, including birth weight, birth centile, small for gestational age (SGA) and spontaneous preterm birth.

## 2. Materials and Methods

### 2.1. Study Design and Participants

We analysed samples from three English centres from the Screening for Pregnancy Endpoints (SCOPE) birth cohort. The SCOPE study was previously reported in detail elsewhere [17]. In brief, the SCOPE study recruited 5690 nulliparous women with singleton pregnancies before 15 weeks’ gestation, between November 2004 and January 2011 in New Zealand, Australia, UK and Ireland. The UK women were recruited in the cities of Leeds, Manchester and London between November 2004 and August 2008. Women were excluded if they presented a high risk of SGA, pre-eclampsia or spontaneous preterm birth [17]. The original SCOPE study protocol has been registered at clinicaltrials.gov NCT02357667.

Participants were interviewed during antenatal clinic visits at 15 (± 1) and 20 (± 1) weeks’ gestation. Participant characteristics were recorded including age, ethnicity, diet, body mass index (BMI), smoking, marital status, employment and socioeconomic status. Participant socioeconomic index was scored using the New Zealand Socio-economic Index 1996. Spot urines were collected for 643 women at the three UK cities; 619 samples were collected at 15 weeks and 585 at 20 weeks, with 547 providing samples at both time points.

### 2.2. Urinary Iodine Measurement

Urinary iodine concentration (UIC) was measured in all samples taken at 15 and 20 weeks of gestation. Assessment of individual iodine status typically requires a series of 24 hour urine collections, however spot urines are considered an acceptable tool for assessing iodine status on a population basis [3]. It is additionally possible to eliminate some inter-individual variation from spot-urine samples by correcting UIC by urinary creatinine concentration (g/L) [18]. We report iodine status results both as the raw iodine concentration (µg/L) and the iodine-creatinine ratio (I:Cr) (µg/g).

Urinary ^127^iodine concentration was measured at the University of Leeds using inductively coupled plasma-mass spectrometry (ICP-MS) (Thermo iCAP Q, Hemel Hempstead, UK), accredited by the Centers for Disease Control and Prevention (CDC) Ensuring the Quality of Urinary Iodine Procedures (EQUIP) standardisation programme. Test samples, calibration standards, and quality control samples were diluted 1:10 prior to analysis. Individual samples consisted of 500 μL of participant urine, 4000 μL of diluent (1% tetramethylammonium hydroxide (Sigma Aldrich), 0.01% Triton X-100 (Sigma Aldrich)) and 500 µL high purity H_2_O (>18.2 MΩ.cm). Tellurium (10 μg/L) (Sigma Aldrich) was included as an internal standard for the ICP-MS analysis. Sample concentration was determined against a urine matrix matched calibration curve spiked with 0, 5, 10, 40, 70, 100, 400 and 800 μg/L iodide (Sigma Aldrich). The accuracy of the results was further validated through the inclusion of internal quality control urines: A (target iodine value: 59.9 μg/L, range: 47.2–72.5), B (98.1 μg/L, 88.3–107.9) and C (158.9 μg/L, 140.2–177.5). External validation of our results was provided by inclusion of the certified reference material Seronorm Trace Metal Urine Level 1 (target value: 105 μg/L, certified iodine range: 84–126), and through participation in the CDC EQUIP programme. Observed values were 63.3 μg/L (*n* = 99), 98.6 μg/L (*n* = 99) and 157.9 μg/L (*n* = 99) for quality control urines A, B and C respectively. The observed value for certified reference material was 102.7 μg/L (*n* = 33). Assessment of intra-run precision gave coefficients of variation (CV) of 1.42% at 60 μg/L, 1.67% at 98 μg/L and 2.34% at 159 μg/L. Inter-run precision gave a CV of 2.4% at 60 μg/L, 4.7% at 98 μg/L and 6.0% at 159 μg/L. The method limit of quantification was 1.46 μg/L.

Urinary creatinine concentrations were assessed through a standard microplate assay utilising the Jaffe reaction. Assessment of creatinine intra-assay precision gave a CV of 2% at 10 mg/L, 1% at 70 mg/L and 1.1% at 120 mg/L. Assessment of inter-assay precision gave a CV of 14.5% at 10 mg/L, 9.7% at 70 mg/L and 5.5% at 120 mg/L.

### 2.3. Outcomes

The primary and a priori outcome for the analysis was birth weight centile. Secondary outcomes were birth weight, SGA, and spontaneous preterm birth. SGA was defined as a birth weight centile <10th on a customised centile chart that accounts for maternal height, weight, parity, ethnicity, neonatal gestation at delivery and sex [19]. This definition was selected due to its standard use in obstetrics and applicability to the UK population, and as it is likely to identify foetal growth restriction. Spontaneous preterm birth was defined as spontaneous preterm labour or preterm premature rupture of the membranes resulting in preterm birth at less than 37 weeks’ gestation. These birth outcomes were selected due to the key role of dietary iodine and thyroid hormones in regulating neonatal growth, and the existence of prior studies identifying links between birth weight and preterm birth [20,21,22,23].

### 2.4. Statistical Analysis

Prior to analysis, a set of exclusion parameters were applied. Analysis was limited to participants with urine samples at both time points, using the mean UIC per participant [24]. Exploratory data analysis was then conducted to ensure all values were plausible and all combinations of values were possible. Two participants had duplicate measures at the same time point removed and an additional four participants were excluded due to possessing UIC values more than 3 standard deviations outside of the mean UIC on a log scale. Participant characteristics were described, stratified by location. Geometric mean and median UIC and I:Cr are presented with 95% confidence intervals (CIs), alongside the proportion of participants with estimated UIC < 50 µg/L. All analyses were completed using Stata version 15.1 (StataCorp. Stata statistical software: Release 15.1. College Station, TX: Stata Corporation, Texas, USA, 2017).

Linear regression was used to quantify associations between UIC and I:Cr for continuous birth outcomes (birth weight and birth weight centile) and logistic regression was used for binary outcomes (SGA and spontaneous preterm birth < 37 weeks). Estimates (with 95% CI) are presented based on the arithmetic mean across the two time points for both UIC and I:Cr for mothers providing both samples (primary analysis) and stratified by time point (secondary analyses). All results are presented as unadjusted and adjusted for known confounders. Directed acyclic graphs were used to inform which variables should be included as covariates, and to prevent over-adjustment [25]. Ethnicity, age, geographic location and socioeconomic index were adjusted for by inclusion in the regression model, except where ethnicity was already accounted for in the birth weight centile and SGA outcomes. Maternal height, weight, parity, neonatal gestation at delivery and sex were also taken into account in the definition of birth weight centile and SGA. The robustness of birthweight and spontaneous preterm birth results to additional adjustment for maternal BMI was assessed.

### 2.5. Ethics Statement

Ethical approval was gained from local research ethics committees for each SCOPE recruitment site (South East Multi-centre Research Ethics Committee/St Thomas Hospital Research Ethics Committee, 2005082; South East Multi-Centre Research Ethics Committee/Central Manchester Research Ethics Committee, 06/MRE01/98). All participants provided written informed consent.

## 3. Results

### 3.1. Participant Characteristics

Table 1 shows participant characteristic summary statistics, including but not limited to maternal age, ethnicity, socioeconomic index, marital status, smoking status, alcohol consumption, height, weight and BMI. The mean maternal age at recruitment was 29 years, 457 (84%) were White Caucasian and 397 (73%) non-smokers (see Table 1). Data were available for 541 participants (151 from London, 260 from Manchester, and 130 from Leeds) after exclusions, with a urine sample available from each participant at both 15 and 20 weeks’ gestation. A total of 1082 urine samples were analysed for iodine, with the overall geometric mean of 130 µg/L and a median of 134 µg/L.

### 3.2. Iodine Status

Geometric mean UIC varied by marital status, geographical location, neonatal gestational age at delivery and gestational time point (Table 2). UIC also varied across time points, with the 20 week concentrations being on average lower than the 15-week concentrations. Participants from Leeds had lower mean UIC (110 μg/L, 95% CI: 99, 121) than participants from Manchester (132 μg/L, 95% CI: 123, 143) or London (145 μg/L, 95% CI: 126, 168). Participants delivering preterm (<37 weeks) generally had similar UIC to those who delivered at term. Consistent with UIC, I:Cr was associated with geographical location and marital status, but not time point or gestational age at delivery. Additionally, I:Cr differed by maternal age category, ethnicity, socioeconomic index and maternal smoking status in the first trimester. Of the three sites, participants at Leeds and Manchester had lower geometric mean I:Cr when compared to London, with Leeds having the lowest. Individuals with UIC (<50 µg/L) constituted 9% (95% CI: 7, 12) across all three sites, with these individuals constituting 9%, 7% and 13% of the Leeds, Manchester and London sites respectively.

### 3.3. Birth Outcomes 

Table 3 shows the associations between mean UIC per participant and continuous birth outcomes. There was little evidence of any association between UIC and birth weight (−14 g per 50 µg/L increase in mean UIC, 95% CI: −35, +6). Results were similar when iodine was corrected for creatinine, with no evidence of any association with birth weight (−8 g per 50 µg/g increase in mean I:Cr, 95% CI: −20, +4). When birth weight was considered further as a customised birth centile, again no evidence of an association was found between mean UIC or mean I:Cr. Birth outcome associations with UIC were largely consistent when iodine levels were assessed at individual time points.

Table 4 shows associations between mean UIC per participant and binary birth outcomes. No evidence of an association was observed between SGA and mean UIC or mean I:Cr. There was no evidence that spontaneous preterm birth was associated with mean iodine concentration, with a 50 µg/L increase in mean UIC associated with 0% increase in the odds of spontaneous preterm birth (95% CI: −16%, +20%), nor was there any evidence found of an association between mean urinary I:Cr and spontaneous preterm birth (odds ratio: 0.91, 95% CI: 0.78, 1.07).

Estimates for birth weight and spontaneous preterm birth outcomes were not substantially changed on adjustment for maternal BMI (data not shown).

## 4. Discussion

World Health Organization (WHO) guidelines are commonly applied for assessing iodine deficiency disorders [3]. These guidelines define a median population UIC > 100 µg/L as iodine sufficient for school-age children and non-pregnant adults. In addition the proportion of participants with a UIC < 50 µg/L must not exceed 20% of the population to be considered iodine sufficient. Currently, WHO guidelines define iodine status during pregnancy as either sufficient (median UIC 150–249 µg/L) or insufficient (<150 µg/L). Each of the three cities we have assessed in the SCOPE birth cohort would be classed as iodine insufficient under these pregnancy guidelines. Median UIC varied between each city with London and Manchester cohorts giving the highest median UIC, whilst Leeds gave the lowest. On average, UIC decreased between 15 and 20 weeks of pregnancy, though no such decrease was observed when assessing I:Cr.

Despite the presence of iodine insufficiency in the cohort, there was no evidence of any association between UIC or I:Cr and birth outcomes. Only a small number of studies have assessed potential links between UIC in pregnancy and neonatal outcomes [13,20,21,22,23], all of which involved populations that were either sufficient [20,23] or borderline sufficient in iodine [13,21,22]. The majority of studies have found no association between UIC and birth weight [13,20,22]. Two studies found positive associations between UIC and birth weight [21,23], but these associations were inconsistent across trimesters.

Charoenratana et al. [20] found that lower UIC was associated with increased odds of preterm birth in Thai women and studies examining links between thyroid hormone concentrations and birth outcomes suggest that pregnant women with hypothyroidism are at greater risk of preterm birth than euthyroid women [26,27]. Whilst these results appear to contradict ours, it should be noted that our study examined gestational time points in the second trimester of pregnancy, whereas Charoenratana et al. measured UIC across all three trimesters and. In addition, a number of other studies have found no evidence of an association between UIC and preterm birth [13,21,22]. Mild iodine deficiency is known to be linked to maternal hypothyroxinemia, a condition characterized by low free T4 levels despite normal free T3 concentrations [28], which in turn has been related to adverse neurodevelopmental outcomes [29]. Conversely, the association between hyperthyroidism and preterm birth is well documented, with the majority of these cases being associated with increased levels of anti-thyroid autoantibodies [27]. However this study was unable to measure thyroid hormone levels, nor examined a hyperthyroid population. We suggest that further studies are warranted in order to clarify the relationship between iodine status and thyroid hormone concentrations with birth outcomes during pregnancy.

Whilst London and Manchester participants had higher UIC values, they also had smaller proportions of participants in the lowest category of socioeconomic status, with London having the smallest. In contrast, Leeds had the highest proportion. As our study found that lower socioeconomic status was associated with lower UIC, our results may suggest that regions of the UK with a greater proportion of citizens with a lower socioeconomic index may be at greater risk of iodine insufficiency. Additionally, participants in the 25–29 age category generally had lower UIC values than participants in other age categories, whilst active smokers generally had lower mean UIC values than non-smokers. Whilst the proportion of participants in each cohort with UIC values <50 µg/L was adequate under WHO guidelines for the general population (<20% of all participants), our observed proportions still constitute a large number of pregnant women who may be at risk for inadequate iodine intakes, if these samples are representative of the UK population as a whole [3].

This study also found that UIC decreased on average between the two time points. A number of changes occur in foetal physiology during the second trimester of pregnancy, with the most notable being the activation of the foetal thyroid between weeks 16–18 of pregnancy [30]. An increase in free T4 allocation to the foetal cerebral cortex also occurs from the mid-point of the first trimester, peaking in weeks 13–20 of pregnancy [30]. These increases in foetal thyroid hormone demand, along with increased iodine trapping and hormone production by the maternal thyroid to meet these demands, could account for the decrease in UIC seen at week 20, compared with week 15. Changes in renal physiology during pregnancy may also have contributed to the observed decrease in UIC [30,31]. Renal iodine clearance via urine increases by roughly 35–50% throughout pregnancy, a change that results in a shortfall in available serum iodine and a compensatory increase of iodine uptake by the thyroid [31]. Previous studies have suggested that the maternal thyroid responds to low iodine status by upregulating thyroid stimulating hormone (TSH) to stimulate increased iodine capture [31]. The decrease in UIC we observe between weeks 15 and 20 may translate into altered TSH, free T3 or T4, but unfortunately this could not be assessed in the present study. A further possibility is that these observed differences were due to differing urine dilutions of participants at these time points. Notably the difference in UIC observed between time points was not observed when using the I:Cr, a correction aimed at reducing inter-individual variability in UIC [22]. This result suggests that iodine excretion may actually remain relatively constant between 15 and 20 weeks.

Our study was limited by a number of factors. Sample collection was restricted to the second trimester of pregnancy, and used spot rather than 24 h urine samples, which are limited in their applicability for assessing iodine status to the population level, rather than providing information on individual chronic iodine intakes. In addition, recruitment was restricted to a small number of UK cities and as a result, the study may have been limited in its assessment of iodine status in surrounding suburban and rural areas. Our study also has a number of strengths. These include the large number of participants available for each of our three geographical sites compared with many prior UK iodine studies, the collection of demographic details through face-to-face interviews with participants, the use of robust analytical procedures for assessing biomarkers of iodine status, and controlling for a number of potentially confounding variables in the statistical modelling. To the best of our knowledge, this is the first study to compare iodine status between geographically separate UK populations. 

## 5. Conclusions

In conclusion, we have demonstrated that the iodine status of pregnant women in our study cohort is generally insufficient by WHO guidelines, with variations in iodine concentrations occurring across different cities in the UK. This finding broadly supports conclusions made by prior UK studies that pregnant women in the UK are insufficient in dietary iodine [10,11,12,13,14,15,16]. However, we have not found evidence that this is adversely associated with the birth outcomes assessed in this study. More detailed follow-up of maternal thyroid function, foetal and childhood growth, plus cognitive development are required to assess the long-term implications of this level of iodine insufficiency.

## Figures and Tables

**Table 1 nutrients-11-00441-t001:** Participant characteristics by location.

Participant Characteristics	Overall (*N* = 541)	London (*N* = 151)	Manchester (*N* = 260)	Leeds (*N* = 130)
	*n* (%)	*n* (%)	*n* (%)	*n* (%)
Mean maternal age (years)				
18–24	121 (22)	12 (8)	62 (24)	47 (36)
25–29	160 (30)	37 (25)	84 (32)	39 (30)
30–34	189 (35)	71 (47)	79 (30)	39 (30)
35+	71 (13)	31 (21)	35 (13)	5 (4)
Maternal ethnicity				
White Caucasian	457 (84)	123 (81)	217 (83)	117 (90)
Other	84 (16)	28 (19)	43 (17)	13 (10)
Maternal socioeconomic index (quartiles)				
Q1 (more deprived)	126 (23)	12 (8)	66 (25)	48 (37)
Q2	133 (25)	31 (21)	68 (26)	34 (26)
Q3	145 (27)	52 (34)	62 (24)	31 (24)
Q4 (less deprived)	137 (25)	56 (37)	64 (25)	17 (13)
Marital status				
Single	84 (16)	14 (9)	38 (15)	32 (25)
Married	258 (48	95 (63	114 (44)	49 (38)
Living as married	199 (37)	42 (28)	108 (42)	49 (38)
Maternal smoking status in 1st trimester				
Non-smoker	397 (73)	128 (85)	186 (72)	83 (64)
Smoker	144 (27)	23 (15)	74 (28)	47 (36)
Maternal alcohol consumption in 1st trimester (units/week)				
0	153 (28)	34 (23)	85 (33)	34 (26)
<2	94 (17)	38 (25)	38 (15)	18 (14)
2–7	160 (30)	41 (27)	79 (30)	40 (31)
>7	134 (25)	38 (25)	58 (22)	38 (29)
Gravidity				
1	385 (71)	105 (70)	184 (71)	96 (74)
2	119 (22)	35 (23)	61 (23)	23 (18)
3+	37 (7)	11 (7)	15 (6)	11 (8)
Maternal height (cm)				
<160	103 (19)	31 (21)	53 (20)	19 (15)
160–169	300 (55)	85 (56)	148 (57)	67 (52)
170+	138 (26)	35 (23)	59 (23)	44 (34)
Maternal weight at 1st visit (kg)				
<60	154 (28)	44 (29)	77 (30)	33 (25)
60–69	190 (35)	60 (40)	89 (34)	41 (32)
70–79	107 (20)	34 (23	48 (18)	25 (19)
80+	90 (17)	13 (9)	46 (18)	31 (24)
Maternal body mass index at 1st visit (kg/m^2^)				
<20	38 (7)	12 (8)	13 (5)	13 (10)
20–25	289 (53)	84 (56)	143 (55)	62 (48)
25–30	148 (27)	42 (28	69 (27)	37 (28)
30+	66 (12)	13 (9)	35 (13)	18 (14)
Spontaneous preterm delivery (<37 weeks)				
No	523 (97)	144 (95)	252 (97)	127 (98)
Yes	18 (3)	7 (5)	8 (3)	3 (2)
Neonatal sex				
Male	277 (51)	78 (52)	140 (54)	59 (45)
Female	264 (49)	73 (48)	120 (46)	71 (55)

Numbers represent the number of individuals in each category, with the percentage in brackets.

**Table 2 nutrients-11-00441-t002:** Urinary iodine concentration and iodine-to-creatinine ratio by participant characteristics.

Participant Characteristics	Mean Urinary Iodine Concentration (µg/L)			Mean Iodine-to-Creatinine ratio (µg/g)	
	Geometric mean (95% CI)	Median (95% CI)	Percent < 50µg/L (95% CI)	Geometric mean (95% CI)	Median (95% CI)
Overall	130 (122, 138)	134 (124, 145)	9 (7, 12)	193 (182, 205)	186 (175, 201)
Maternal age (years)					
18–24	127 (114, 141)	135 (117, 154)	9 (5, 16)	137 (123, 153)	145 (126, 157)
25–29	116 (104, 129)	122 (100, 135)	9 (5, 14)	190 (171, 211)	186 (162, 222)
30–34	138 (124, 154)	140 (119, 164)	11 (7, 16)	228 (208, 251)	219 (201, 254)
35+	147 (122, 176)	144 (122, 191)	6 (2, 14)	232 (197, 272)	210 (175, 249)
Maternal ethnicity					
White Caucasian	130 (122, 139)	134 (124, 147)	10 (7, 13)	201 (189, 214)	190 (176, 209)
Other	127 (112, 143)	124 (103, 147)	4 (1, 10)	155 (135, 178)	160 (142, 193)
Maternal socioeconomic index (quartiles)					
Q1 (More deprived)	125 (112, 139)	128 (109, 163)	6 (2, 11)	157 (141, 174)	153 (144, 182)
Q2	132 (117, 149)	135 (114, 150)	9 (5, 15)	199 (177, 223)	199 (161, 225)
Q3	133 (118, 151)	135 (113, 162)	10 (6, 16)	209 (186, 235)	202 (172, 224)
Q4 (Less deprived)	129 (114, 146)	132 (119, 149)	11 (6, 17)	209 (186, 236)	200 (177, 228)
Marital status					
Single	112 (98, 128)	114 (99, 135)	13 (7, 22)	140 (123, 159)	150 (127, 175)
Married	128 (117, 141)	130 (119, 146)	10 (7, 15)	216 (199, 234)	211 (186, 230)
Living as married	140 (128, 154)	147 (126, 162)	6 (3, 10)	192 (174, 211)	184 (163, 211)
Maternal smoking status in 1st trimester					
Non-smoker	134 (124, 144)	135 (124, 151)	9 (7, 13)	207 (193, 221)	200 (180, 217)
Smoker	119 (108, 132)	126 (111, 143)	8 (4, 14)	160 (145, 177)	154 (143, 180)
Maternal alcohol consumption in 1st trimester (units/week)					
0	133 (120, 148)	135 (117, 161)	7 (4, 12)	183 (166, 202)	184 (163, 212)
<2	124 (107, 143)	125 (99, 153)	11 (5, 19)	193 (168, 222)	186 (161, 206)
2–7	135 (121, 151)	132 (119, 155)	6 (3, 11)	205 (183, 230)	201 (169, 225)
>7	124 (108, 141)	134 (108, 158)	13 (8, 20)	191 (169, 216)	178 (158, 203)
Gravidity					
1	129 (120, 139)	134 (124, 149)	10 (7, 13)	198 (185, 211)	188 (174, 209)
2	131 (117, 147)	125 (115, 151)	7 (3, 13)	183 (161, 209)	183 (147, 210)
3+	130 (104, 163)	120 (102, 174)	8 (2, 22)	179 (148, 218)	188 (154, 217)
Maternal height (cm)					
<160	129 (112, 148)	126 (104, 159)	8 (3, 15)	202 (175, 234)	175 (157, 212)
160–169	134 (123, 145)	134 (123, 152)	10 (7, 14)	196 (182, 212)	197 (176, 212)
170+	122 (110, 137)	131 (115, 152)	9 (5, 15)	180 (161, 202)	170 (152, 206)
Maternal weight at 1st visit (kg)					
<60	131 (116, 146)	126 (112, 152)	9 (5, 15)	207 (185, 231)	201 (175, 234)
60-69	124 (112, 137)	125 (111, 141)	8 (5, 13)	195 (177, 214)	190 (170, 209)
70–79	130 (112, 149)	135 (114, 172)	12 (7, 20)	202 (177, 231)	177 (159, 209)
80+	143 (125, 163)	157 (127, 173)	8 (3, 15)	160 (140, 184)	159 (128, 210)
Maternal body mass index at 1st visit (kg/m^2^)					
<20	131 (106, 163)	130 (98, 175)	5 (1, 18)	178 (138, 229)	157 (121, 253)
20–25	127 (116, 138)	127 (117, 146)	10 (7, 14)	207 (192, 224)	200 (181, 221)
25–30	128 (115, 143)	134 (115, 151)	9 (5, 15)	186 (167, 209)	177 (161, 205)
30+	149 (128, 174)	164 (126, 182)	8 (3, 17)	162 (139, 190)	161 (132, 210)
Geographical location					
London	145 (126, 168)	130 (114, 177)	13 (8, 20)	242 (210, 279)	228 (181, 273)
Manchester	132 (123, 143)	139 (126, 158)	7 (4, 10)	183 (171, 196)	183 (165, 202)
Leeds	110 (99, 121)	116 (99, 135)	9 (5, 16)	165 (149, 183)	168 (149, 193)
Spontaneous preterm delivery (<37 weeks)					
No	130 (122, 138)	134 (124, 144)	9 (7, 12)	194 (183, 206)	188 (175, 202)
Yes	132 (89, 196)	147 (75, 254)	11 (1, 35)	171 (129, 226)	155 (140, 246)
Neonatal sex					
Male	128 (118, 139)	134 (119, 147)	10 (7, 14)	199 (184, 215)	186 (174, 210)
Female	132 (121, 143)	134 (122, 151)	8 (5, 12)	187 (172, 204)	184 (163, 206)
Appointment					
15 weeks gestation	134 (125, 143)	135 (127, 152)	12 (10, 15)	176 (165, 188)	172 (159, 189)
20 weeks gestation	101 (94, 109)	107 (95, 115)	23 (19, 27)	188 (177, 200)	183 (170, 200)

**Table 3 nutrients-11-00441-t003:** Change in continuous birth outcomes associated with a 50 µg/L increment in mean urinary iodine concentration or 50 µg/g increment in iodine-to-creatinine ratio. 95% confidence intervals are included in brackets.

		Change in Outcome per 50 µg/L Increment in Urinary Iodine Concentration		Change in Outcome per 50 µg/g Increment in Iodine-to-Creatinine Ratio	
	Appointment	Unadjusted	Adjusted *	Unadjusted	Adjusted *
Birth weight (g)	15 weeks	−6 g (21, +10)	−7 g (−23, +8)	0 g (−9, +9)	−4 g (−14, +5)
	20 weeks	−9 g (−27, +9)	−13 g (−31, +5)	−3 g (−14, +7)	−8 g (−19, +4)
	Mean	−10 g (−30, +10)	−14 g (−35, +6)	−2 g (−14, +9)	−8 g (−20, +4)
Birth weight centile **	15 weeks	0.0% (−0.8, +0.8)	−0.2% (−0.9, +0.6)	+0.1% (−0.3, +0.6)	0.0% (−0.5, +0.4)
	20 weeks	+0.1% (−0.8, +1.0)	−0.2% (−1.1, +0.7)	0.0% (−0.5, +0.5)	−0.2% (−0.7, +0.4)
	Mean	0.0% (−0.9, +1.0)	−0.2% (−1.3, +0.8)	+0.1% (−0.5, +0.6)	−0.1% (−0.7, +0.5)

* Adjusted for maternal age, city, ethnicity and socioeconomic index. ** Additionally, gestation at delivery, maternal height and weight, parity and child’s sex were taken into account in the definition of birth weight centile.

**Table 4 nutrients-11-00441-t004:** Relative increase in odds of binary birth outcomes (odds ratios) associated with a 50 µg/L increment in mean urinary iodine concentration or 50 µg/g increment in iodine-to-creatinine ratio. 95% confidence intervals are included in brackets.

		Odds Ratio per 50 µg/L Increment in Urinary Iodine Concentration		Odds Ratio per 50 µg/g Increment in Iodine-to-Creatinine Ratio	
	Visit	Unadjusted	Adjusted *	Unadjusted	Adjusted *
Small for gestational age **	15 weeks	1.03 (0.96, 1.10)	1.05 (0.97, 1.13)	1.00 (0.95, 1.04)	1.01 (0.97, 1.06)
	20 weeks	1.05 (0.97, 1.13)	1.08 (0.99, 1.16)	1.00 (0.95, 1.05)	1.02 (0.97, 1.07)
	Mean	1.05 (0.97, 1.14)	1.09 (1.00, 1.20)	1.00 (0.94, 1.05)	1.02 (0.96, 1.09)
Spontaneous preterm birth (<37 weeks)	15 weeks	1.03 (0.90, 1.18)	1.02 (0.89, 1.17)	0.96 (0.85, 1.09)	0.96 (0.85, 1.08)
	20 weeks	0.98 (0.82, 1.17)	0.97 (0.81, 1.17)	0.89 (0.75, 1.07)	0.89 (0.76, 1.06)
	Mean	1.02 (0.85, 1.22)	1.00 (0.84, 1.20)	0.91 (0.77, 1.08)	0.91 (0.78, 1.07)

* Adjusted for maternal age, city, ethnicity and socioeconomic index. ** Additionally, gestation at delivery, maternal height and weight, parity and child’s sex were taken into account in the definition of small-for-gestational-age.

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
