# Peer review of "Maternal Iodine Status and Associations with Birth Outcomes in Three Major Cities in the United Kingdom"

_nutrients, 2019, doi:10.3390/nu11020441_

Reviewer 1 Report

This is a multicenter UK cohort study assessing urinary iodine concentrations during gestation in association with birth weight and other obstetric outcomes.

Introduction, lines 71-74: it would be helpful to explain here why these specific birth outcomes were selected for analysis.

Discussion, lines 263-4, "our observed proportions still constitute a large number of pregnant women with inadequate iodine concentrations": having two spot urine concentrations below the WHO threshold for population medians may men women are at risk for inadequate intakes, but is not diagnostic of iodine deficiency. This sentence could be re-worded to make this clearer, and the fact that spot UCI values cannot be used as a biomarker for chronic iodine intakes should be further discussed in the paragraph on study limitations (lines 290-294).

Discussion, lines 275-6, "a compensatory change to iodine clearance by the thyroid" would be clearer as "  a compensatory increase of iodine uptake by the thyroid."

Author Response

Response to Reviewer 1

Introduction, lines 71-74: it would be helpful to explain here why these specific birth outcomes were selected for analysis.

 We have updated this section as requested detailing the selection process for the specific birth outcomes. Specifically these birth outcomes were selected due to a combination of thyroid hormones being involved in regulating correct growth and development, and the existence of prior studies identifying potential links between iodine deficiency and birth weight, SGA and pre-term birth.

 Discussion, lines 263-4, "our observed proportions still constitute a large number of pregnant women with inadequate iodine concentrations": having two spot urine concentrations below the WHO threshold for population medians may men women are at risk for inadequate intakes, but is not diagnostic of iodine deficiency. This sentence could be re-worded to make this clearer, and the fact that spot UCI values cannot be used as a biomarker for chronic iodine intakes should be further discussed in the paragraph on study limitations (lines 290-294).

 We agree with the reviewer that this section could benefit from clarification as outlined above. We have now revised the statement (Lines 270-271) and added further detail on this point to the discussion (Lines 301-303).

 Discussion, lines 275-6, "a compensatory change to iodine clearance by the thyroid" would be clearer as "a compensatory increase of iodine uptake by the thyroid."

 The sentence has been changed as recommended.

 We thank the reviewer for their comments.

Reviewer 2 Report

Dear Authors,

     I have read with huge interest your manuscript entitled “Maternal iodine status and associations with birth outcomes in three major cities in the United Kingdom” to be considered for publication in Nutrients.

This is an interesting study in terms of Public Health: data regarding three different locations within the UK, a large number of participants and a powerful statistical analysis (taking into account potential confounders). Moreover, these findings support the epidemiological fact that iodine status among British pregnant women remains insufficient.

 However, I have some comments to make:

1.       It would be interesting to remark in the introduction that iodine requirements increase from early stages of gestation. In fact, there is a body of evidence concerning the consequences of iodine deficiency during the first half of pregnancy, which are particularly linked to fetal neurodevelopment, so they may be irreversible.

2.       Therefore, the outcomes to assess the impact of iodine deficiency throughout pregnancy cannot be constrained to the time of birth. In this regard, I believe that certain statements such as “we have not found evidence that this is associated with adverse birth outcomes” (lines 299-300) might be misunderstood.

3.       From these results we can figure out an interesting profile of iodine-deficient pregnant women (young, smokers, from low socioeconomic background), who should be targeted for supplementation or other preventive approaches. Please, consider to highlight this clinical/social profile in the discussion. I agree with the authors that the proportion of participants with UIC values < 50 µg/L is concerning (lines 261-264).

4.       Line 270. It would desirable to add some information regarding the well-known causal relationship between iodine deficiency and maternal hypothyroxinemia as well as its consequences on neurodevelopment.

5.       Finally, my personal interpretation of changes in UIC and I:Cr is quite different from that offered by the authors. Renal iodine clearance increases throughout pregnancy, and contributes to make detectable underlying iodine deficiencies. So, when using I:Cr you will find an increase (compare your results with those from reference 16), but the raw UIC will be lower at later gestational ages. In fact, your results are consistent with a leakage of iodine through urine.

Author Response

Response to Reviewer 2

It would be interesting to remark in the introduction that iodine requirements increase from early stages of gestation. In fact, there is a body of evidence concerning the consequences of iodine deficiency during the first half of pregnancy, which are particularly linked to fetal neurodevelopment, so they may be irreversible.

 We agree with the reviewer that this is an important piece of information to highlight. We have included a statement to this effect in the revised manuscript (line’s 50-51).

 Therefore, the outcomes to assess the impact of iodine deficiency throughout pregnancy cannot be constrained to the time of birth. In this regard, I believe that certain statements such as “we have not found evidence that this is associated with adverse birth outcomes” (lines 299-300) might be misunderstood.

 We have reworded this statement to refer specifically to the measures we examined in this study.

 From these results we can figure out an interesting profile of iodine-deficient pregnant women (young, smokers, from low socioeconomic background), who should be targeted for supplementation or other preventive approaches. Please, consider to highlight this clinical/social profile in the discussion. I agree with the authors that the proportion of participants with UIC values < 50 µg/L is concerning (lines 261-264).

 We have added an additional statement to the discussion (Lines 266-268) detailing this information.

 Line 270. It would desirable to add some information regarding the well-known causal relationship between iodine deficiency and maternal hypothyroxinemia as well as its consequences on neurodevelopment.

 As requested we have added an additional statement to the discussion outlining the link between iodine and maternal hypothyroxinemia (Lines 253-255).

 Finally, my personal interpretation of changes in UIC and I:Cr is quite different from that offered by the authors. Renal iodine clearance increases throughout pregnancy, and contributes to make detectable underlying iodine deficiencies. So, when using I:Cr you will find an increase (compare your results with those from reference 16), but the raw UIC will be lower at later gestational ages. In fact, your results are consistent with a leakage of iodine through urine.

 We entirely agree with the reviewer that the decrease in raw UIC may be due to increased renal clearance (resulting in increased “leakage” of iodine through urine later in pregnancy). We apologise if our interpretation was not clear and have amended it for greater clarity being more explicit about the loss of iodine via increased renal iodine clearance later in pregnancy.

 We thank the reviewer for their comments.
